# Mapping mitonuclear epistasis using a novel recombinant yeast population

Tuc H. M. Nguyen[1,2], Austen Tinz-Burdick[1], Meghan Lenhardt[1], Margaret Geertz[1], Franchesca Ramirez[1], Mark Schwartz[1], Michael Toledano[1], Brooke Bonney[1], Benjamin Gaebler[1], Weiwei Liu[1], John F. Wolters[1], Kenneth Chiu[3], Anthony C. Fiumera[1], Heather L. Fiumera[1]*

**1** Department of Biological Sciences, Binghamton University, Binghamton, New York, United States of America, **2** Department of Biological Sciences, New York University, New York, New York, United States of America, **3** Department of Computer Sciences, Binghamton University, Binghamton, New York, United States of America

* hfiumera@binghamton.edu

**Data Availability Statement:** Reads for the MNRC strains can be retrieved from NCBI under the BioProject ID PRJNA871925.Scripts, SNP table

## Abstract

Genetic variation in mitochondrial and nuclear genomes can perturb mitonuclear interactions and lead to phenotypic differences between individuals and populations. Despite their importance to most complex traits, it has been difficult to identify the interacting mitonuclear loci. Here, we present a novel advanced intercrossed population of *Saccharomyces cerevisiae* yeasts, called the Mitonuclear Recombinant Collection (MNRC), designed explicitly for detecting mitonuclear loci contributing to complex traits. For validation, we focused on mapping genes that contribute to the spontaneous loss of mitochondrial DNA (mtDNA) that leads to the *petite* phenotype in yeast. We found that rates of *petite* formation in natural populations are variable and influenced by genetic variation in nuclear DNA, mtDNA and mitonuclear interactions. We mapped nuclear and mitonuclear alleles contributing to mtDNA stability using the MNRC by integrating a term for mitonuclear epistasis into a genome-wide association model. We found that the associated mitonuclear loci play roles in mitotic growth most likely responding to retrograde signals from mitochondria, while the associated nuclear loci with main effects are involved in genome replication. We observed a positive correlation between growth rates and *petite* frequencies, suggesting a fitness tradeoff between mitotic growth and mtDNA stability. We also found that mtDNA stability was correlated with a mobile mitochondrial GC-cluster that is present in certain populations of yeast and that selection for nuclear alleles that stabilize mtDNA may be rapidly occurring. The MNRC provides a powerful tool for identifying mitonuclear interacting loci that will help us to better understand genotype-phenotype relationships and coevolutionary trajectories.

## Author summary

Mitochondrial functions require genes from nuclear and mitochondrial genomes that must work together. These interactions influence organismal fitness and coevolutionary processes yet it is difficult to identify the genes involved. Here, we created a novel

and data are available at https://github.com/mito32/Mitonuclear-Recombinant-Collection.

**Funding:** This research was supported by an NIH award (GM101320) to HLF, ACF and KC that provided salaries to THMN, ML, JFW, KC, ACF and HLF. MS was supported by an NSF REU (EEC 1757846). ATB, BB, MT, and BG received Binghamton University Undergraduate Research awards. The funders had no role in study design, data collection and analysis, decision to publish, or preparation of the manuscript.

**Competing interests:** The authors have declared that no competing interests exist.

collection of yeast designed explicitly for mapping mitonuclear genes. We used this collection to reveal genes influencing the maintenance of mitochondrial DNAs (mtDNAs), a trait important for human health. The mapping population presented here is an important new resource that will help to understand genotype-phenotype relationships and coevolutionary trajectories. Additionally, this work provides insight into mechanisms underlying mtDNA stability.

## Background

Interactions between mitochondrial and nuclear genomes are essential for the mitochondrial functions that power eukaryotic life. Mitonuclear interactions can be direct, as physical contacts between mitochondrial and nuclear genes and their products are needed for mitochondrial DNA (mtDNA) replication and maintenance and transcription, translation, assembly and function of mitochondrially encoded components of oxidative phosphorylation (OXPHOS) complexes [1]. Mitonuclear interactions can also be indirect, though anterograde (nucleus-to-mitochondria) and retrograde (mitochondria-to-nucleus) signaling where metabolites, biochemicals or RNAs direct gene expression in response to metabolic needs or environmental stressors [2,3]. Genetic variation can alter the efficiencies of these interactions leading to phenotypic differences between individuals and populations.

Mitonuclear epistasis, defined as the non-additive phenotypic effects of interacting mitochondrial and nuclear allele pairs, has been demonstrated across Eukarya, including vertebrates [4–6], invertebrates [7–12], plants [13] and fungi [14–21]. In humans, allelic variation in mitonuclear interactions contributes to human diseases [22–26]. Mitonuclear epistasis occurs within populations [27,28], between populations of the same species [15,29–35] and between closely related species [17,36], suggesting that physiologically-relevant mitonuclear epistasis is ubiquitous in natural populations. Mitonuclear loci are clearly influencing phenotypes that shape the structure of natural populations and are important for understanding evolutionary and coevolutionary processes, including speciation. Identifying the mitonuclear interactions that contribute to complex traits is challenging and largely a goal unmet in biology.

Selection for mitonuclear interactions may contribute to speciation [29,37]. Because of the large interest in uncovering speciation loci, strategies to identify mitonuclear epistasis often focus on analyzing mitonuclear incompatibilities in interspecific and inter-population hybrids or in mitonuclear hybrids where nuclear genomes from one population are paired with the mtDNAs from another. Sometimes candidate genes for these incompatibilities can be revealed through deductive reasoning. For example, in crosses between populations of *Tigriopus californicus* where mtDNA inheritance was controlled, F2 hybrids showed mitonuclear-specific OXPHOS enzyme activities and mtDNA copy number differences, prompting investigations into mitochondrially encoded electron transport proteins and the mtRNA polymerase [38,39]. In *Drosophila*, a mtDNA from *D. simulans* paired with a nuclear genome from *D. melanogaster* resulted in a mitonuclear genotype with impaired development and reproductive fitness [40]. Fortuitously for mapping purposes, the mtDNA sequences in this mitonuclear panel had very few polymorphisms, enabling the causative alleles behind this incompatibility (a mitochondrially encoded tRNA and a nuclear encoded tRNA synthetase) to be identified [36].

Other genetic approaches can also be used to reveal mitonuclear epistatic loci. Chromosomal replacements in interspecific *Saccharomyces* yeast hybrids, followed by plasmid library screening, allowed the identification of mitonuclear incompatibilities between nuclear genes

encoding intron splicing factors from one species and their mitochondrially encoded targets in the other [17,41,42]. Quantitative trait loci (QTL) mapping approaches using genotype-phenotype associations of recombinant progeny containing different mtDNAs have also been used to seek intraspecific mitonuclear incompatibilities [43–46]. Analysis of meiotic segregants following forced polyploidy enabled the detection of QTLs for interspecific mitonuclear incompatibilities contributing to sterility barriers in *Saccharomyces* [47]. Due to the low resolution of QTL mapping, specific loci were not identified, but in some cases, regions of mitonuclear genomic interest were implicated. An association study in wild yeast isolates identified mitochondrial variants of *ATP6* that associated with sensitivity to the ATP synthase inhibitor, oligomycin [20]. When mtDNAs with these alleles were introduced into iso-nuclear heterozygous genetic backgrounds, oligomycin sensitivity was sometimes dependent on the nuclear background in addition to the *ATP6* allele, suggesting mitonuclear epistasis. The interacting nuclear loci were not identified. (Interestingly, these isonuclear diploids also demonstrated non-respiratory phenotypes that were dependent on both mitotypes and nuclear genotypes, suggesting pervasive mitonuclear epistasis [20]). Other approaches to uncovering mitonuclear epistatic loci include differential expression analysis [5,48–50] and experimental evolution [51,52]. Because relatively few genetic backgrounds are used for most mapping approaches, even if single gene pairs are identified, it is not clear if these approaches will reveal a general picture of mitonuclear epistasis or lineage specific idiosyncrasies.

Previously, we showed that statistical estimates of mitonuclear epistasis explained over 30% of the phenotypic variances observed in a panel of *S. cerevisiae* yeasts consisting of 225 unique mitonuclear genotypes [15]. In the current study, our goal was to map naturally occurring alleles leading to mitonuclear epistasis in yeast populations. Detecting mitonuclear epistatic loci (or any g x g interaction) using association approaches can be challenging due to low allele frequencies, the large number of tests required to detect pairwise epistasis, dominance effects, and the potential for the environment to affect genetic interactions [37,50,53,54]. To overcome some of these challenges, we created a multiparent advanced intercrossed recombinant population of *S. cerevisiae* haploids designed explicitly to detect mitonuclear epistasis through association testing. Inheritance of mtDNA was controlled, resulting in a mapping collection where each nuclear genotype is paired with up to three different mitotypes. This should increase statistical power to detect mitonuclear interactions across the full mapping population, detect nuclear effects within a given mitotype and provide finer mapping resolution. Mitonuclear interactions could be integrated into phenotype-genotype association models allowing detection of loci contributing to complex traits.

To test our novel mapping strategy, we focused on mapping naturally-occurring alleles that contribute to the stability of the mitochondrial genome. Large scale deletions within mtDNAs are a common form of mtDNA instability, leading to mitochondrial heteroplasmies (defined as having more than one mitotype present) and deterioration of organismal health [55–59]. *Saccharomyces* yeasts do not tolerate mtDNA heteroplasmies and will fix for a single mitotype after just a few mitotic divisions [60]. Yeast cells fixed for these truncated mtDNAs can grow via fermentation and will form small, *petite*, colonies in comparison to the larger *grande* colonies formed by respiring cells, making it possible to quantify rates of mtDNA deletions [61]. Laboratory strains have accumulated multiple genetic variants leading to increased *petite* frequencies [62]. Mitochondrial research was heralded through curiosity of genetic causes of the *petite* phenotype in yeast [63]. Detailed mechanistic studies have, over decades, revealed a large number of genes contributing to mitochondrial respiration [[64] and refs therein], including those involved in replication and maintenance of mtDNAs [62,64–67]. This long list of genes is likely incomplete, given that studies typically focus on laboratory strains in singular, controlled environments. While these genes all interact with the mitochondria, it is not

obvious which exhibit specific mitotype effects. Additionally, these studies are unable to provide insight into the potential for natural selection to be operating. We reasoned that natural genetic variation would lead to differences in *petite* frequencies among wild isolates of *S. cerevisiae* through mitonuclear epistasis and that our recombinant population would enable the identification of mitonuclear epistatic loci.

Here, we show that mtDNA stability in wild *S. cerevisiae* yeasts is influenced by mitonuclear epistasis and by independent effects of both nuclear and mitochondrial genomes. We describe the construction of the Mitonuclear Recombinant Collection, and how we were able to use this to detect nuclear loci that participate in the mtDNA stability through main (independent) effects and mitonuclear interactions. We identified mitonuclear epistatic loci involved with mitotic growth, correlating rates of mitotic growth with mtDNA stability. This novel strategy and mitonuclear recombinant strain collection provide new tools for identifying mitonuclear loci that are important in nature.

## Results

### mtDNA stability is a quantitative trait influenced by mitochondrial GC-clusters and mitonuclear interactions

Given the mechanistic complexities of mtDNA replication and maintenance, we hypothesized that standing genetic variation would contribute to differences in mtDNA stability. To test this, we examined the frequencies of spontaneous *petite* colony formation in 21 isolates of *S. cerevisiae* (**Fig 1** and **S1 Table**). Rates of mtDNA deletions in these isolates ranged from 0.3 to 9.9%. All 21 isolates had lower *petite* frequencies than S288c, a widely used laboratory strain known to harbor nuclear variants that promote exceptionally high rates of *petite* formations [62]. Within the wild isolates, genetic variation within and between broadly-defined ancestral populations contributed to differences in the rates of mtDNA loss ($P<0.001$ for strain and population, **S2 Table**). Strains with Wine/European or mosaic origins had the lowest rates of *petite* formation, while strains with West African or Sake/Asian origins had the highest.

The differences in *petite* formation rate between wild isolates could be due to genetic variation in mtDNAs, nuclear genomes and/or mitonuclear interactions. To examine the independent effects of different mitotypes on mtDNA stability, we introduced 18 different mtDNAs into a common nuclear background (Y55) and assayed rates of mtDNA deletions. In these isonuclear strains, *petite* frequencies ranged from 5.5 to 31.0% due to differences in mitotypes (ANOVA, $P<0.001$, **S3 Table**). To determine if *petite* formation correlated with any particular mitochondrial feature, we analyzed the mtDNAs for their length, intron content, GC% and GC-cluster content (**S4 Table**). The GC-rich regions of yeast mtDNAs are predominantly due to short (~30–40 bp) mobile GC-rich palindromic sequences called GC-clusters that interrupt long intergenic AT-rich sequences [71,72]. GC content and GC-cluster numbers were tightly correlated ($r = 0.95$, $P<0.001$), and both were correlated with the total length of mtDNAs ($r = 0.59$, $P = 0.001$ and $r = 0.70$, $P<0.001$, respectively). *Petite* frequencies, however, did not correlate with the total length of mtDNAs ($r = 0.12$, $P = 0.63$) nor the total lengths of intron sequences ($r = -0.09$, $P = 0.72$). Thus, it is likely that GC-rich regions play an important role in mtDNA stability. In fact, we found that *petite* frequencies correlated with overall GC% of mtDNAs (Pearson's $r = 0.59$, $P = 0.013$, **Fig 2A**) but not the total numbers of GC-clusters ($r = 0.44$, $P = 0.07$), mtDNA length ($r = 0.12$, $P = 0.63$) nor total length of intron sequences ($r = -0.09$, $P = 0.72$). A presumed mechanism for *petite* formation is illegitimate recombination in mitochondrial GC-clusters [61,73,74]. To see if any particular type of GC-cluster could explain mtDNA instabilities, we sorted the GC-clusters into 9 classes described by sequence homologies [75]. We found that the M4 GC-cluster class, but no other, correlated with *petite*

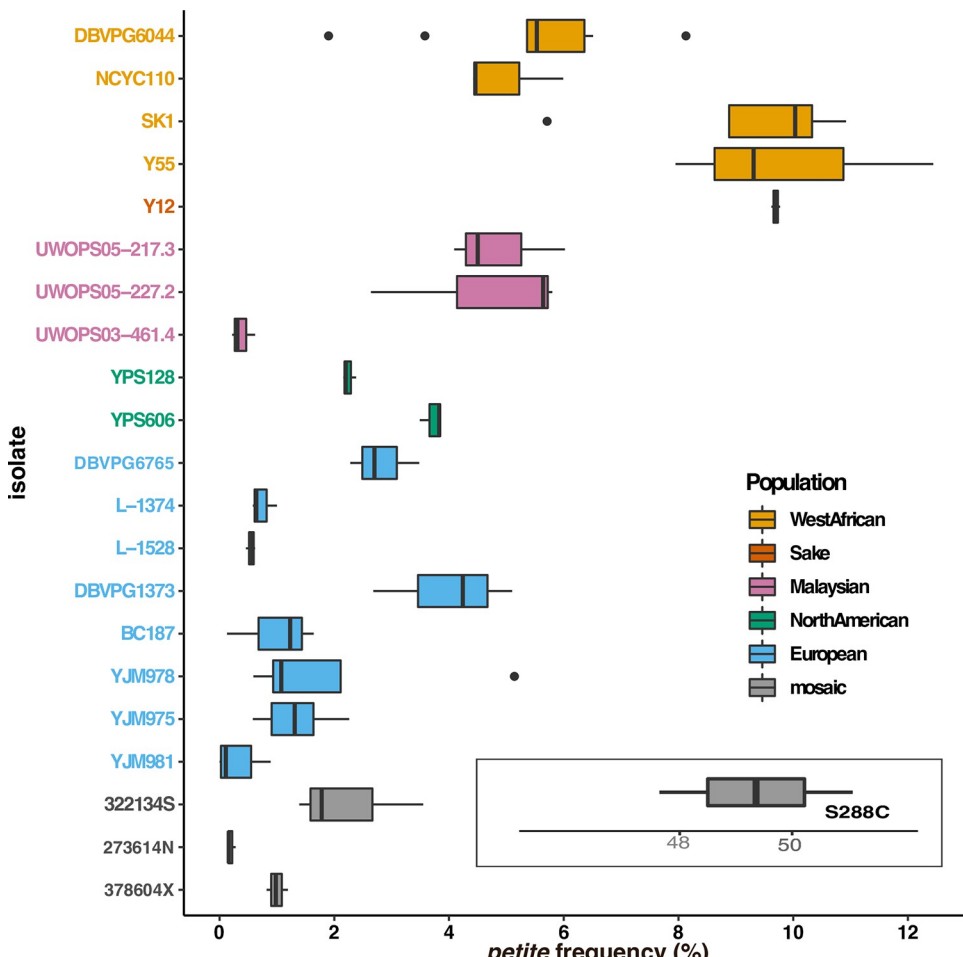

**Fig 1. Genetic variation contributes to *petite* frequency.** Boxplots showing *petite* frequencies (total *petite* colonies/ total colonies *100) are presented for haploid derivatives of 21 wild *S. cerevisiae* isolates and the reference strain, S288c. Strains are colored by broad population identities from [68–70] as indicated in the key. The *petite* frequency for the reference strain is offset for scaling purposes. *Petite* frequencies differ between populations and between strains within populations (ANOVA, **S2 Table**).

frequencies (r = 0.67, *P* = 0.004, **Fig 2B**). This particular cluster appears to be expanding in the mtDNAs of strains with West African lineages [76] and these strains explain the observed correlation. It is also possible that an unidentified feature in the West African mtDNAs contributes to their instability.

GC-clusters (or some other West African mtDNA features) do not solely control rates of *petite* formation. When a mtDNA from the West African strain with the highest *petite* frequency was introduced into different nuclear backgrounds, *petite* frequencies were increased in 4 of 8 nuclear backgrounds tested and decreased in 2 of them (**Fig 2C**). This is consistent with both nuclear and mitonuclear effects. Nuclear backgrounds from Wine/European origins maintained relatively low *petite* frequencies even when harboring this GC-cluster-rich mtDNA, indicating that nuclear genotypes largely control mtDNA stability, at least for these strains.

Some of the synthetic mitonuclear combinations led to even higher *petite* frequencies than observed in the original isolates suggesting that mitonuclear interactions also play a role in mtDNA stability. To formally test for mitonuclear effects on mtDNA stability, we examined

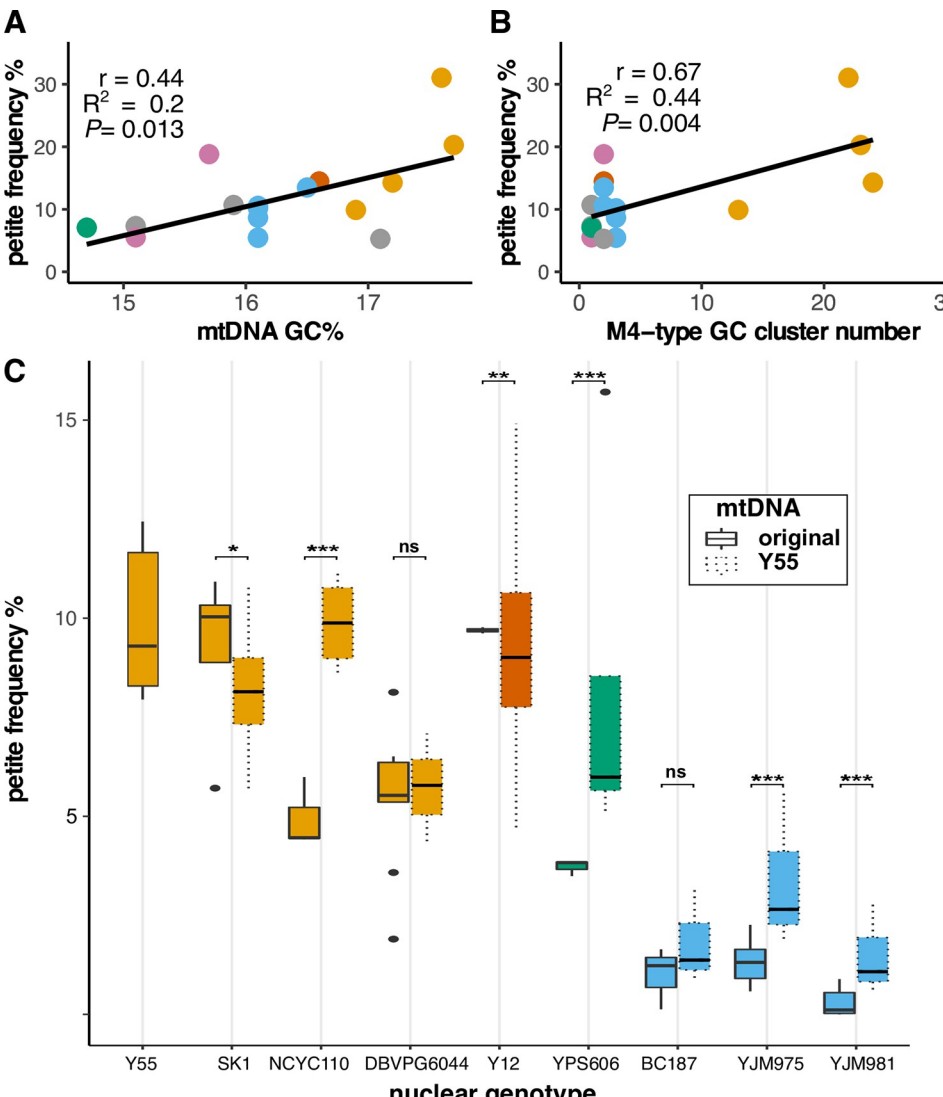

**Fig 2. Mitochondrial GC content influences stability of mtDNAs.** *Petite* frequencies of iso-nuclear strains (containing the nuclear genome from strain Y55) correlate with **A.** total GC% of mtDNAs and **B.** the numbers of M4-type GC-clusters. Pearson's correlation coefficient (r), coefficient of determination ($R^2$) and *P*-value significance are shown. **C.** *Petite* frequencies for strains containing original (solid outlines) or the GC-cluster-rich mtDNA from Y55 (dotted outlines) are shown as box and whisker plots. Colors indicate nuclear genotype population as described in **Fig 1**. Significance of individual ANOVAs comparing the original and synthetic mitonuclear genotypes are shown. * *P<0.05,* ** $P \leq 0.005,$ *** $P \leq 0.001.$

*petite* frequencies in 16 mitonuclear cybrids created by exchanging mtDNAs between 4 strains with West African lineages (4 nuclear x 4 mtDNAs). A two-way ANOVA showed that mtDNA stability was influenced by nuclear genotypes, mitotypes, and mitonuclear interactions (*P*<0.001 for each term, **S5 Table**), where certain mitonuclear combinations showed very large increases in the rates of *petite* formation (**Fig 3**). Within these strains, the original mito-nuclear genotypes had lower *petite* frequencies than any of their synthetic mitonuclear combi-nations (**S1 Fig**). Given the population-specific expansion of the destabilizing M4 clusters within the West African strains, this observation suggests that selection for mitonuclear inter-actions that stabilize mtDNAs has occurred quickly in ways that are strain-specific.

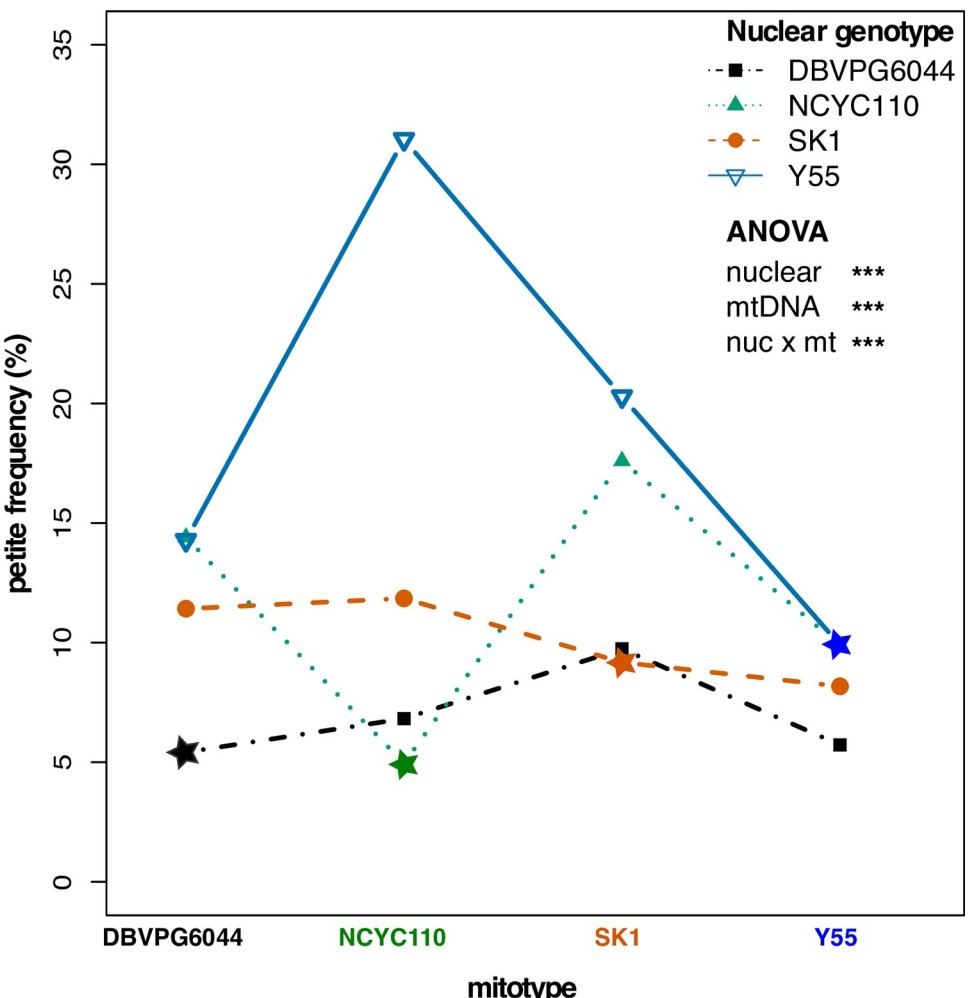

**Fig 3. Coadapted mitonuclear interactions stabilize mtDNAs.** *Petite* frequencies for strains containing 4 nuclear and 4 mtDNA backgrounds are shown as an interaction plot. Colored lines connect a nuclear genotype paired with different mitotypes. The original mitonuclear genotype combination is starred. The nonparallel lines indicate mitonuclear epistasis. ANOVA revealed highly significant nuclear, mtDNA, and mitonuclear genotypes contributions (**S5 Table**). *** $P \leq 0.001$.

## Creation of a Mitonuclear Recombinant Collection for association studies

Mitonuclear interactions explain a significant proportion of phenotypic variances in *S. cerevisiae* yeasts and involve numerous, as yet unmapped, loci [14,15,77]. We sought a general genome-wide mapping approach that would facilitate the mapping of both the nuclear and mitonuclear loci underlying complex traits such as mtDNA stability. We created a multiparent recombinant collection of *S. cerevisiae* strains specifically designed for association mapping of nuclear and mitonuclear loci (called the Mitonuclear Recombinant Collection, or MNRC) (**Fig 4**). To do this, we first replaced the mtDNAs in 25 wild divergent yeast isolates such that each contained an identical mitotype, and then mated them to create each possible heterozygous diploid. The diploids were sporulated and ~10,000 haploid F1 haploid progeny were isolated and then randomly mated. F1 diploids were then isolated and sporulated. Only one laboratory strain (W303, derived from the reference strain) was included as a parent strain. We noted that very few progeny after one round of meiosis contained the selectable markers

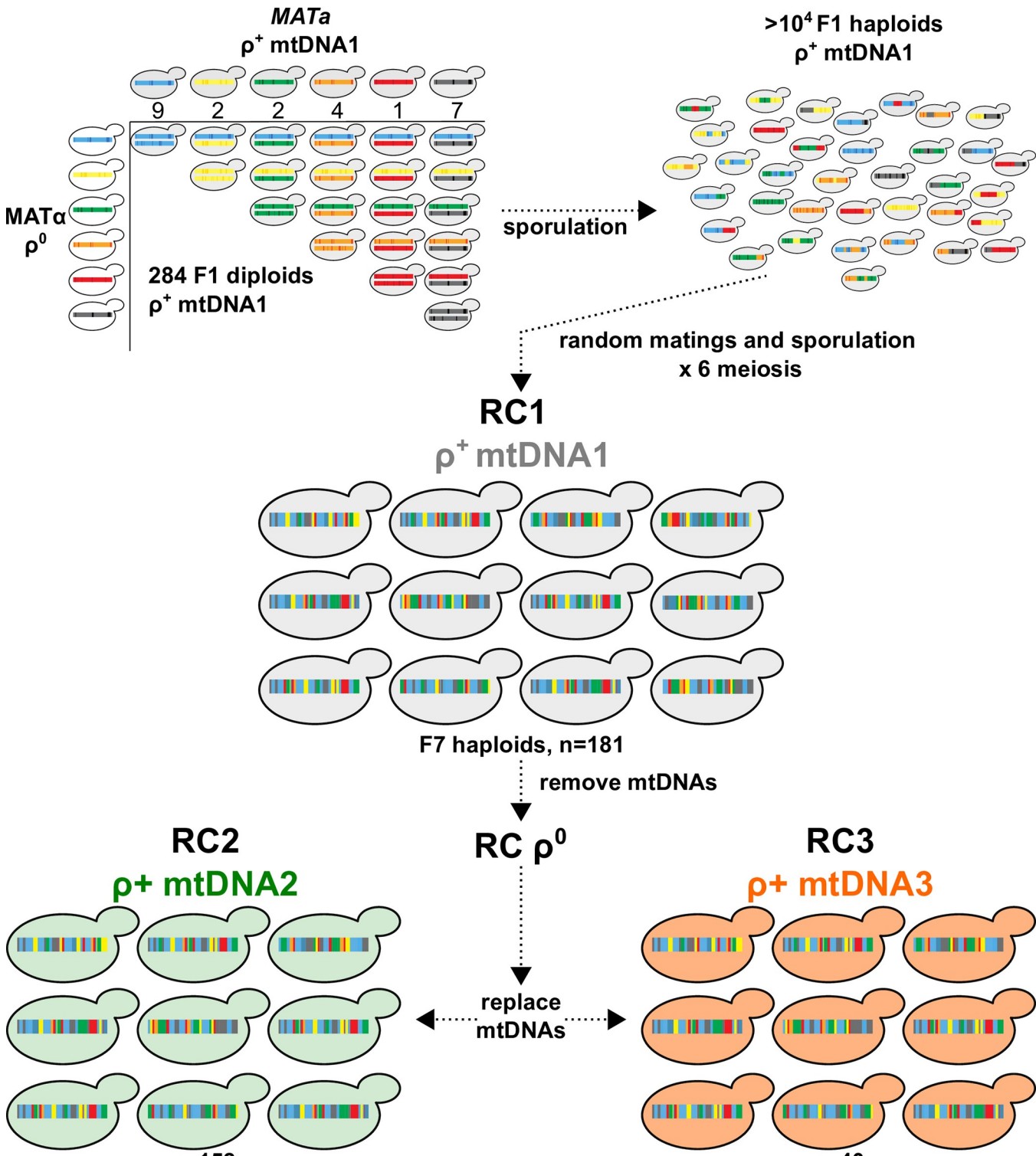

**Fig 4. A Mitonuclear Recombinant Collection designed for association mapping.** 25 unique genetic backgrounds of *S. cerevisiae* fixed for a single mitotype were systematically crossed to create each possible F1 heterozygous diploid with identical mtDNAs (284 total). The numbers of unique genotypes from each parental ancestral background are indicated; blue (Wine/European); yellow (Malaysian); green (North American); orange (West African); black (mosaic). The F1 diploids were sporulated and F1 recombinant haploid progeny isolated. Following 6 rounds of random matings, haploid F7 progeny were isolated to create RC1. The mtDNAs from RC1 were removed (RC1$\rho^0$) and replaced with 2 different mtDNAs, creating RC2 and RC3. The mtDNAs in RC1,2, and 3 are from the wild isolates 273614N, YPS606, and NCYC110, respectively.

found in this strain, suggesting that this genotype was quickly lost from the collection. The process was repeated for a total of 7 rounds of meiosis. A collection of 181 F7 haploids (named Recombinant Collection (RC) 1 or RC1) was isolated and fully sequenced. The mtDNAs from RC1 strains were removed (creating RCρ$^0$) and replaced with two additional mtDNAs via karyogamy-deficient matings, creating populations RC2 and RC3. The GC-cluster content of the mtDNAs in the MNRC are classified as low (117 clusters in RC2), medium (137 clusters in RC1), or high (203 clusters in RC3).

To generate SNPs tables for association testing, RC1 strains were sequenced to ~40x coverage. Paired-end reads were aligned to the *S. cerevisiae* reference genome and the locations of nuclear SNPs and small indels were extracted from each alignment. Polymorphic sites were filtered by removing telomeric regions and SNPs/indels with low allele frequencies (MAF <5%). Following filtering, 24,955 biallelic sites across the 16 yeast chromosomes with an average of ~2200 SNPs/chromosome were available for association testing. Chromosomal polymorphic data are summarized in **S6 Table**. Our read alignments and subsequent analyses did not account for chromosomal rearrangements, such as copy number variants, translocations and inversions that would map to similar locations of the reference genome nor genomic regions absent in the reference strain.

We validated that this novel recombinant population could be used for simple association studies. Strains from RC1 were phenotyped for growth on copper sulfate and an association test was performed to identify SNP variants associated with copper tolerance. A single peak on Chr. 8 coincided with a region containing *CUP1*, the copper binding metallothionein (**S2 Fig**). Variation at this locus is known to lead to high copper tolerances found in Wine/European isolates [78] and has previously been identified through association studies using wild isolates [68,79]. Thus, the recombinant collection produced here is successful for association studies despite a relatively low number of parental strains.

## Nuclear and mitonuclear associations for mtDNA stability

To map nuclear and mitonuclear associations, *petite* frequencies were collected for each strain in RC1, RC2 and RC3. In RC1, the *petite* frequencies ranged from 0.0% to 27.7% forming a continuum, as would be expected for a complex trait involving numerous loci (**Fig 5A**). The same rank orderings were not observed in RC2 or RC3, revealing the influences of mitonuclear interactions. RC3, containing the GC-cluster-rich mtDNA, had a higher average *petite* frequency than RC1 or RC2 (**S3 Fig**).

In theory, the recombinant genomes and fixed mitotypes of the RC strains should reduce effects of population structure, improve statistical power while using a smaller number of samples, limit false positives, and control for mitonuclear interactions. We performed association testing to identify nuclear loci that had both a main effect and interacted with the mtDNA to influence mtDNA stability. Mating types, auxotrophic markers and residual population structure as determined by principal component analyses were included as covariates (see METHODS). The significance profiles of associations for nuclear variants that were independent (*nuclear SNP*, **Fig 5B**) and dependent on mitotype (*nuclear SNP× mtDNA*, **Fig 5C**) were different, providing confidence that the association model is able to detect nuclear features that are unique to either main or epistatic effects.

Nuclear SNPs whose effects were independent of mitotype resulted in stronger associations than mtDNA-dependent alleles. This is not surprising given that independent contributions of nuclear genotypes influence growth phenotypes to a greater extent than mitonuclear interactions [14,15,77]. At a false discovery rate (FDR) < 0.1% (Q-value = 0.001), we observed 130 mtDNA-independent associated SNPs located within or 250 bp upstream of coding sequences

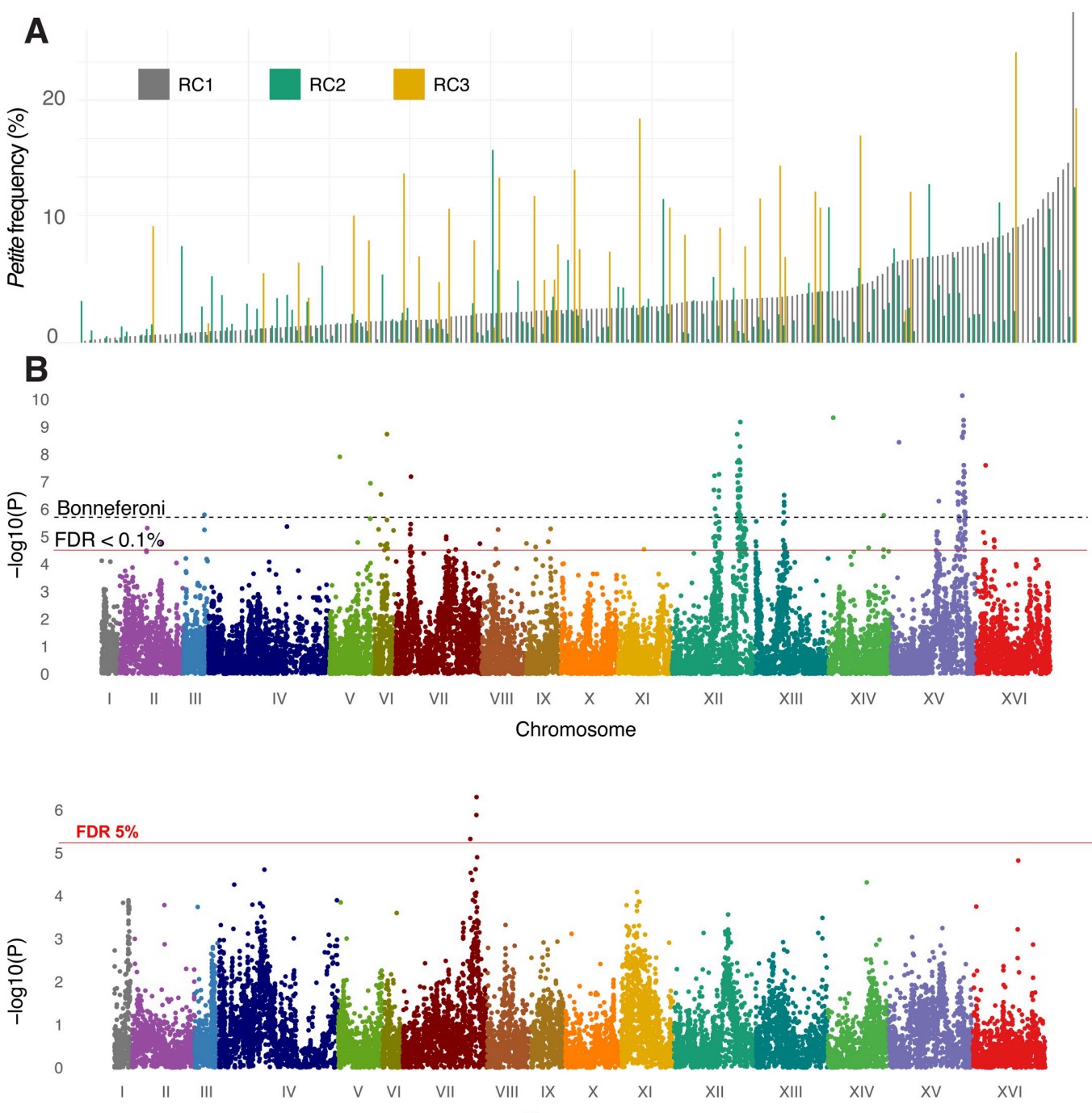

**Fig 5. Nuclear SNPs associate with mtDNA stability through main effects and mitonuclear interactions. A**. *Petite* frequency is a quantitative trait influenced by natural genetic variation. *Petite* frequencies for strains from RC1 (gray) were rank ordered from lowest to highest. The *petite* frequencies for strains from RC2 (green) and RC3 (gold) are plotted next to isonuclear strains from RC1. **B.** Manhattan plots show main effect nuclear SNPs associated with *petite* frequencies in ways that were independent of mtDNA and **C.** nuclear SNPs whose association is dependent on mitotype (ie. mitonuclear). The different plot profiles indicate that the main effect nuclear SNPs are different that those involved with mitonuclear interactions. FDR thresholds at 0.1% ($P < 4.1 \times 10^{-5}$) for nuclear associations and 5.0% ($P < 1.2 \times 10^{-5}$) for mitonuclear associations and a conservative Bonferroni threshold ($P < 2.0 \times 10^{-6}$) are shown.

(S7 Table). In comparison, at FDR <5% (Q-value = 0.05), there were 3 mtDNA-dependent associated SNPs (S8 Table). Strong nuclear effects could mask mitonuclear interactions. We identified the alleles with the strongest effects by calculating the effect size of each SNP with mitotype-independent associations as the difference between the average *petite* frequencies of each allele weighted by its frequency in the recombinant collections (S9 Table). This revealed that the highest effect size was attributed to a SNP on Chr. 15 predicting a previously uncharacterized G50D missense mutation in *MIP1*, the mitochondrial DNA polymerase required for replication and maintenance of mtDNA. To improve power of detecting mitonuclear associations, we repeated the analysis including the *MIP1* SNPs as covariates. This removed the mitotype-independent associations on Chr. 15 and increased the numbers of significant mitonuclear associations from 3 to 27 without changing the overall association profiles (S4 Fig and S8 Table).

## Mitonuclear associations: Mitotic growth signaling pathways

The 27 mitonuclear SNP associations corresponded to 21 unique genes, including 11 in a QTL on Chr. 7 (S8 Table). Within this QTL, the three strongest associations corresponded to SNPs near the coding start sites of the genes *HGH1* and *SMI1* and an in-frame deletion within *BNS1*. Interestingly, mitochondrial activities have not been shown for these genes. A second QTL on Chr. 1 contained a SNP upstream of *YAT1*. To verify the involvement of these genes in mitonuclear interactions affecting mtDNA stability, we first looked at how removing each gene influenced *petite* frequencies. In the parental background of the *S. cerevisiae* knockout collection (BY4741) and with the BY4741 mitotype, the null mutant *hgh1Δ* lowered rates of *petite* formation while null mutants *bns1Δ*, *smi1Δ* and *yat1Δ* showed no significant effect (Fig 6A).

We next introduced two different mitotypes (the GC-cluster-rich mitotype NCYC110 used in RC3 and the GC-cluster-poor mitotype YPS606 used in RC2) into the parental and deletion strains and measured *petite* frequencies. The mitonuclear interactions detected in our GWAS model may be dependent on particular nuclear backgrounds. Still, mitonuclear interactions in the reference strain background (BY4741) were readily observed; the GC-cluster-rich mitotype led to higher *petite* frequencies in the *hgh1Δ* and *yat1Δ* deletion strains, whereas the same mitotype led to a lower *petite* frequency in the parental background (Fig 6B). This is in contrast with the mitotype independent effect observed in *est1Δ* strains, whose *petite* frequencies roughly paralleled the parental background. Two-way ANOVAs showed highly significant mitonuclear interactions when comparing the parental strain to *hgh1Δ* and *yat1Δ* deletion strains with each mtDNA comparison (P<0.001, S10 Table). Similar mitonuclear effects were also observed for *bns1Δ* and *smi1Δ* when compared to the parental background (S5 Fig and S11 Table). While the high *petite* frequency of the parental strain complicates the interpretation of these assays in terms of understanding how these physiological interactions influence population level phenotypic variation, the differential responses provide strong evidence that the association model was successful in identifying genes that influence mtDNA stability through mitonuclear epistatic interactions.

We tested if the mitonuclear associations could be explained by differences in gene expression. mRNAs were extracted from a subset of 18 strains from the GC-cluster-rich RC2 and GC-cluster-poor RC3 collections. We attempted to control for *MIP1* alleles and the 2 most common SNP haplotypes for mitonuclear candidate genes on Chr. 12 (containing *HGH1*, *BNS1*, *and SMI1)* and Chr. 1 (containing *YAT1*). Across the subset of RC2 and RC3 strains containing alternate mitonuclear haplotypes with each *MIP1* allele, *YAT1* SNPs (Chr. 1) are in complete linkage disequilibrium (LD) with the associated SNPs on Chr. 12 and are thus considered together. Expression of *HGH1* showed a modest positive correlation with *petite*

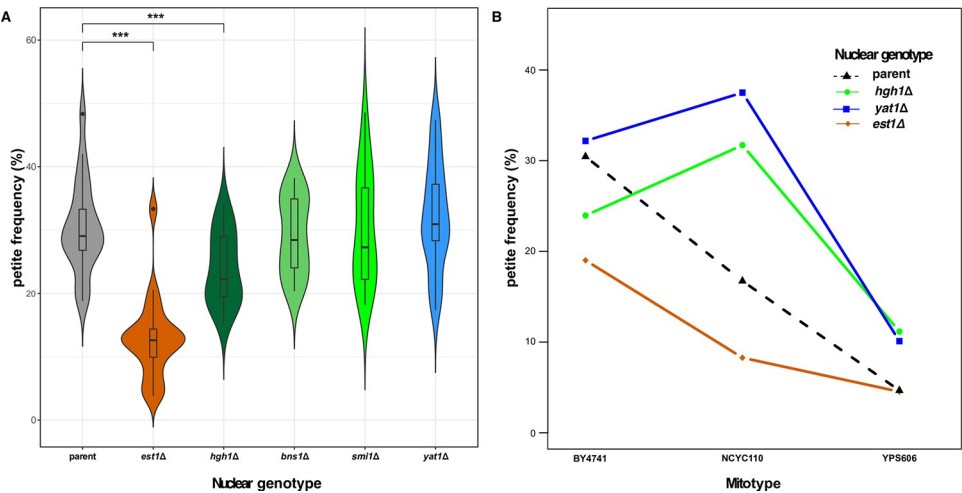

**Fig 6. HGH1 and YAT1 demonstrate physiological mitonuclear interactions for mtDNA stability. A**. *Petite* frequencies for strains lacking genes with mitotype-independent (*est1Δ*) and mitonuclear associations (*hgh1Δ*, *bns1Δ*, *smi1Δ* and *yat1Δ*) are shown as boxes in violin plots. Significant differences between the parental and each deletion strain are shown. * *P<0.05*, ** *P ≤ 0.005*, *** *P ≤ 0.001*. **B**. *hgh1Δ* and *yat1Δ petite* frequencies are dependent on mitonuclear interactions while *est1Δ petite* frequencies are largely mitotype-independent. Colored lines connect each nuclear genotype paired with different mtDNAs. Non-parallel lines indicate that mitotypes influence mtDNA stability through mitonuclear interactions. ANOVA comparisons of the parental strain vs. each deletion strain with any 2 mtDNAs are shown in **S10 Table**. Mitotype BY4741 is the parental mtDNA in the yeast deletion collection. Mitotypes NCYC110 and YPS606 were used in RC3 and RC2, respectively. A minimum of 20 replicates for all *petite* assays were performed.

frequencies (r = 0.43, P = 0.08) (**S6 Fig**). Expression of *MIP1*, *YAT1*, *BNS1* and *SMI1* showed slight but non-significant positive correlations (**S6 Fig**). In these strains, mitotypes did not influence expression of the mitonuclear candidate genes nor on the expression of the mtDNA polymerase, *MIP1* (**S12 Table**). The expression of *MIP1* was, however, influenced by the hap-lotype of the Chr. 12/Chr. 1 QTLs (**S7A Fig**) while expression of *BNS1* was influenced by *MIP1* alleles (**S7B Fig** and **S12 Table**). *BNS1* and *YAT1* expression showed *MIP1* haplotype x mito-nuclear haplotype effects. Interestingly, the higher expression in these genes were observed in strains containing the *MIP1* alleles that led to a high *petite* frequency. These data suggest that the genotypes of *MIP1* and the mitonuclear candidate genes may influence the expression of each other to affect mtDNA stability.

The mitonuclear candidate genes, *YAT1*, *HGH1*, *BNS1*, and *SMI1* are broadly connected to mitotic growth. We observed a significant positive correlation between the growth (measured as colony sizes on solid media) of RC1 and RC2 strains with their *petite* frequencies (r = 0.16, P<0.002, **Fig 7A**). Removal of the one outlier did not influence this correlation. This effect is strongly influenced by strains with the low *petite* frequency allele of *MIP1* at position 943237 (*MIP1*-C) paired with the RC2 mitotype (r = 0.37, P<0.001, **Fig 7B**). *Petite* frequencies of strains with the high *petite* frequency allele (*MIP1*-T) did not show a significant correlation with growth (P>0.05).

To investigate a potential relationship between mitotic growth and mtDNA stability, we compared the growth rates of six wild yeast isolates in different environmental conditions and their corresponding *petite* frequencies (**Fig 7C**). Growth rates of each strain increased between growth in media containing low (0.2%) and high (2.0%) glucose concentrations and high tem-perature (**Fig 7C**, top panel). The increases in Vmax were greatest between 30C and 37C. *Petite* frequencies (bottom panel) also showed increases between 30C and 37C. Overall, conditions

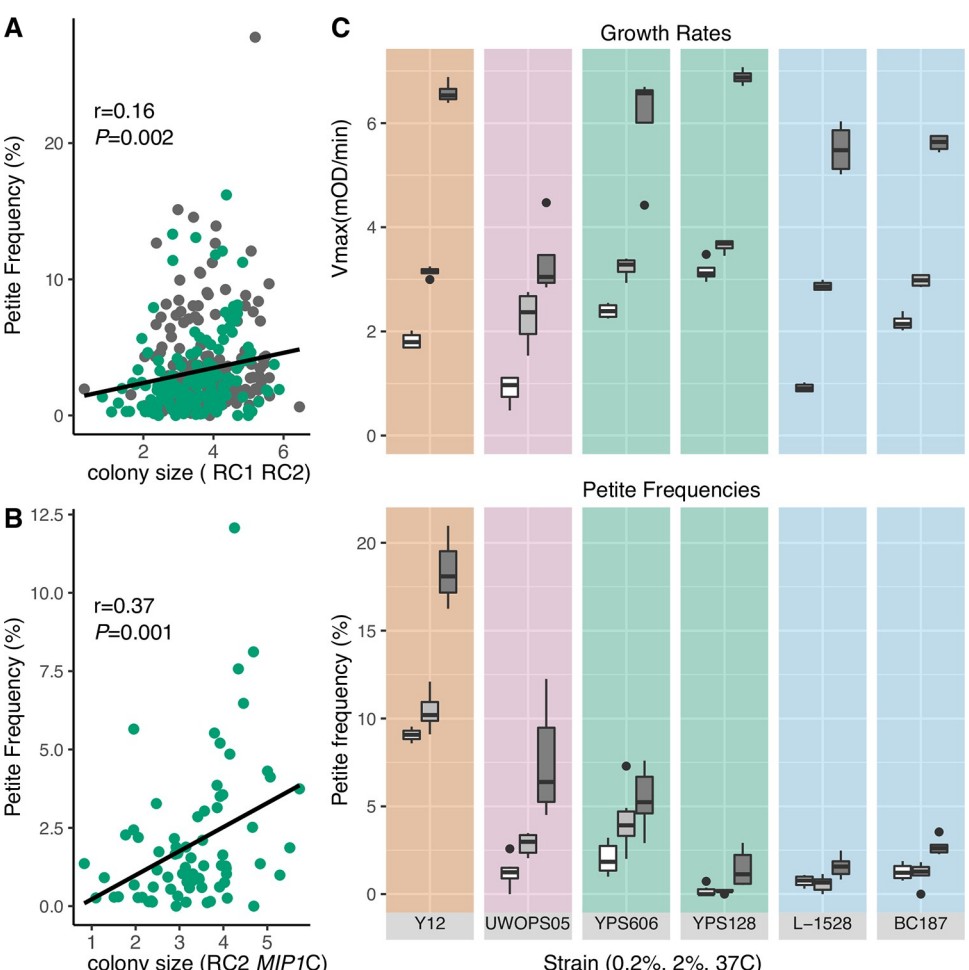

**Fig 7. Rapid mitotic growth increases rates of mtDNA loss.** Growth of **A**. RC1 and RC2 strains and **B**. RC2 strains with the low *petite* frequency *MIP1* allele (943237C) are positively correlated with spontaneous *petite* frequencies. Colony sizes (arbitrary units) are reported as differences in colony size formation on solid media. **C**. Growth rates and *petite* frequencies increased as a result of increased glucose concentration and temperature. Growth rates (top panel) and *petite* frequencies (bottom panel) were measured for 6 strains in media containing low 0.2% (white boxes) or 2% (medium gray boxes) glucose at 30C and in 2% glucose at 37C (dark gray boxes). Shaded backgrounds indicated parental populations as in **Fig 1**. Vmax was estimated from regression lines modeled from growth curves in liquid media.

that increased growth rates increased *petite* frequency, although the *petite* frequency increases were not necessarily proportional to Vmax increases. The two strains with Wine/European ancestries maintained low *petite* frequencies even at high temperature. It is likely that ancestral polymorphisms in these strains have strong nuclear effects that stabilize mtDNAs. Still, these data suggest that increasing mitotic growth rates can lead to a decrease in mtDNA stability, possibly exposing a fitness tradeoff between rapid cell growth and deletions in mtDNA. We also noted that the increase in *petite* frequencies of the West African strains with synthetic mitonuclear genotypes (**S1 Fig**) coincided with an increase in growth rates (**S8 Fig**). Perhaps selection in these strains has favored mitonuclear variants that lower overall growth in order to preserve mtDNA integrity.

## Mitotype-independent associations: mtDNA polymerase and a G-quadruplex stabilizing protein

In addition to identifying mitonuclear loci, our model identified 301 SNPs whose associations were independent of mitotype. Of these, 130 SNPs were located within 250 bp upstream of coding sequences for 86 unique genes (**S7 Table**). Gene Ontology (GO) analyses revealed enrichments in mitochondrially-targeted genes (27 genes), genes involved in mitochondrial organization (9 genes) and homeostatic processes (10 genes) (**S13 Table**). Sixteen of the associated genes generate non-respiratory phenotypes when deleted from the genome [64,80–82]. In all, 33 (38.4%) of the mitotype-independent associations have known mitochondrial activities.

A large number of the associated SNPs occurred within QTLs on chromosomes 12, 13 and 15 (**Fig 5B**) and were likely associated through linkage. Effect sizes (**S9 Table**) were used to help identify causative loci. In addition to the *MIP1* G50D missense mutation in the mitochondrial DNA polymerase, MIP1, two additional *MIP1* missense variants (T540M and H541N) were also significant. These alleles are different from the previously characterized *MIP1-661A* allele that leads to high *petite* frequency in the reference strain [62]. High effect sizes were also seen in associated SNPs in genes flanking *MIP1*, including *FRT1* and *PDR10*, and essential genes *KRE5* and *ALA1*. Deletions of *FRT1* or *PDR10* did not significantly alter *petite* frequencies suggesting that these associations are the result of linkage to *MIP1* (**S9 Fig**).

Thirty-two associated genes were located within two QTLs on Chr. 12. Here, the largest effect size was for a putative missense variant (L13I) in *EST1* (**S9 Table**), encoding a telomerase accessory protein that helps to stabilize G-quadruplex structures in telomeres during certain stages of the cell cycle [83]. We found that a deletion of *EST1* significantly reduced *petite* frequencies as compared to the parental strain in the yeast knockout collection (**Fig 6A**) while a deletion of *TOP3*, a candidate gene tightly linked to *EST1*, did not (**S9 Fig**). This suggests that *EST1* variants likely drive the associations in this QTL. In the parental background of the deletion collection, *petite* frequencies of an *est1Δ* were similarly reduced when different mitotypes were introduced (**Fig 6B**), as would be expected for mitotype-independent associations. Two-way ANOVAs comparing *est1Δ* and parental strains with different mtDNAs showed very strong nuclear and mtDNA effects ($P < 2.2 \times 10^{-16}$) but weak mitonuclear effects ($P < 0.05$, not significant after Bonferonni correction, **S10 Table**). These weak mitonuclear effects are likely driven by low *petite* frequencies in the non-parental mitotypes (especially the YPS606 mitotype). As *petite* frequencies approach zero, differences between nuclear genotypes will appear non-additive.

A second QTL on Chr. 12 contained associations for 5 genes with known mitochondrial functions. *ILV5* is a nonspecific mtDNA binding protein involved in the packaging of mtDNA into nucleoids and is required for mtDNA stability [84]. *NAM2*, a mitochondrial tRNA synthetase [85], and *SSQ1*, a mitochondrial chaperone [86], are required for growth via mitochondrial respiration. We found that a deletion of *MDM30*, a component of the ubiquitin ligase complex involved in mitochondrial morphology and turnover [81], lowered *petite* frequencies while a deletion of *ATG33*, a protein involved in mitochondrial mitophagy [87], had no effect (**S9 Fig**). A single QTL on Chr. 13 contained associations for 6 genes, four of which encode mitochondrially targeted proteins. *COX14* and *SAM37*, and *SOV1* are required for mitochondrial respiration [64,80,81] and *RNA14* is essential for growth. Deletions of a 5ᵗʰ gene, *CSM3*, did not significantly alter *petite* frequencies (**S9 Fig**).

## Discussion

### Mitonuclear Recombinant Collection

We created a multiparent advanced intercross population (Mitonuclear Recombinant Collection or MNRC) to specifically identify mitonuclear interactions driven by allelic variation in *S. cerevisiae* yeasts through association testing. The MNRC incorporates the genetic variation from 25 wild isolates into ~200 fully sequenced recombinant strains. By avoiding most laboratory strains in the mating design, the potential to identify genetic variants that are relevant to natural systems is improved. These strains were paired with three different mitotypes, enabling the detection of mitonuclear interactions via genome-wide association tests. Using the MNRC, we were able to identify loci contributing to mtDNA stability through independent (main) effects and mitonuclear interactions.

The MNRC was designed to help overcome many of the challenges of detecting epistatic interactions. Multiple rounds of random mating should lower existing LD created by the strong population structure in *S. cerevisiae* [68,70] and reduce the numbers of strains needed for association testing. The MNRC strains are haploid, eliminating dominance effects. Because novel mtDNAs created through mitochondrial recombination may produce unexplained phenotypic variances [19,77], mtDNA inheritance was controlled. By specifically pairing each mitotype with the nuclear recombinants, the power to detect specific mitonuclear loci was increased. Yeasts do not maintain mtDNA heteroplasmies and can be cryogenically preserved such that allele frequencies of the full collection are not subject to sampling drift. Thus, unexplained phenotypic variance should be reduced and statistical power increased. Environments can easily be controlled in yeast, eliminating unexplained variances due to mtDNA x nuclear x environment interactions. We showed that the MNRC can successfully detect nuclear alleles that contribute to phenotypes through independent (main) or epistatic (mitonuclear) effects.

We did not test for nuclear-nuclear epistasis across the recombinant collections. Allele frequency differences due to the different numbers of strains in each mitotype population, or novel mutations during strain construction, could theoretically falsely identify nuclear-nuclear interactions as a mitonuclear interactions. Of the three SNPs with the most significant mitonuclear associations (Chr7 positions 949775, 951886 and 871605), the largest allele frequency difference between mitotype populations was 12.4% (S15 Table). We looked for evidence of nuclear x nuclear epistasis at these SNPs by performing additional association analyses using RC1 (*petite frequency ~ nuclear SNP$_i$ * mitonuclear-associated SNP$_j$*). We found no evidence of nuclear x nuclear epistasis at these loci (with the lowest FDR = 0.08). Balancing the strain numbers in each mitotype population and testing for nuclear x nuclear interactions for candidate SNPs will reduce the likelihood of misidentifying mitonuclear interactions in future studies.

### Loci affecting mtDNA stability

**Mitonuclear interactions.** Our model analyzing *petite* frequencies found epistatic associations for SNPs in four genes (*HGH1*, *SMI1*, *BNS1*, and *YAT1*) that are broadly related to mitotic growth. Independent (main) effects had much higher effect sizes than mitonuclear effects and so main effects may mask physiological effects in any given background. Consistent with this, the correlation between colony size formation (a proxy for mitotic growth) and *petite* frequencies in RC1 and RC2 increased when *MIP1* alleles were taken into account (**Fig 7A**). Even though the genetic background of the reference strain contains many alleles that lead to increased *petite* frequencies [62], we were able to confirm that these statistical epistatic associations conferred physiological mitonuclear epistasis by showing that null mutations of genes with mitonuclear associations showed mitotype-specific effects as compared to the parental

background and to a null mutation of a gene with mitotype-independent associations. *YAT1* encodes a mitochondrial outer membrane protein that participates in mitochondrial metabolism by importing the acyl groups that enter the Krebs cycle for energy production [88,89]. Industrial yeast strains have likely adapted to high levels of toxic aldehydes produced during ethanol production by increasing *YAT1* copy numbers [90]. The Yat1 protein is phosphorylated [91], suggesting that its activity is regulated through cell signaling. *HGH1* encodes a translation factor chaperone involved in protein synthesis that is activated in response to DNA damage [92,93]. The molecular function of *SMI1* is unknown, but it is thought to be a member of the cell signaling pathway that regulates cell wall biosynthesis during mitotic growth [94]. Null mutations of *SMI1* have reduced respiratory growth [95], suggesting its activity influences mitochondrial function. During mitosis, *BNS1* participates in the signaling network directing the exit from anaphase [96]. A high-quality mitochondrial proteomics screen found the Bns1 protein in the mitochondrial matrix [97]. Possibly, these genes are involved in mitonuclear epistasis by changing growth parameters in response to retrograde signals produced by mitochondria with damaged mtDNAs. We expect this to be a general trend. Dominant mutations that suppressed the low growth phenotype of strains with mitochondrial ribosomal protein defects also led to higher *petite* frequencies [98].

Mitotic growth also linked mitonuclear epistasis and mtDNA stability when exchanging mtDNAs between strains from West African lineages. When mtDNAs were exchanged between West African isolates, strains with original, coadapted mitonuclear genome combinations had lower *petite* frequencies (**Fig 3**) and mitotic growth rates (**S8 Fig**) than strains harboring introduced, non-coadapted, mtDNAs. This suggests that selection for mtDNA-stabilizing mitonuclear alleles is rapid and may come at the cost of lowering overall growth rates and may even dictate an upper limit for optimal growth. It is interesting to note that mtDNA instability is a hallmark of fast growing human cancer cells [99]. Experimental evolutions aimed at altering growth rates may be one way to demonstrate this potential fitness trade-off. In yeast cells exposed to oxidative stress agents, partial deletions within mtDNAs may initially be under genetic control, as a way to quickly reduce endogenous ROS levels by preventing OXPHOS [67]. We found that growth rates and *petite* frequencies were increased by altering environmental conditions (**Fig 7B**). It is possible the increased OXPHOS requirements of rapidly dividing cells may also stimulate this retrograde signaling activity, resulting in higher rates of mtDNA deletions.

In constant environments, there should be rapid selection for variants providing optimal growth rates. This selection may occur more often in yeast where cells are likely to experience singular environments over a generation time. There is some evidence for this. When grown in media attempting to emulate natural habitats, we previously showed that coadapted mitonuclear genome combinations had higher growth rates [15]. Because there is a wide landscape of potential alleles involved in growth, different coadapted mitonuclear loci may evolve rapidly in different populations. In yeast, this could be important in maintaining allelic variation across the species. It will be interesting to see whether growth is a general feature of mitonuclear loci contributing to phenotypes other than mtDNA stability. Given that mitochondria are central to maintaining cellular homeostasis, this may be likely.

**Main effects.**   Loci with main effects for mtDNA stability detected by our model interact with mtDNA through non-specific binding. In the MNRC, missense alleles of the mtDNA polymerase, *MIP1*, predicting G50D, T540M and H541N amino acid changes, had the strongest effects on *petite* frequencies. It is important to note that only one relative of the yeast reference strain was included in the parental collection of the MNRC, and that selection against some of the auxotrophic markers in this genotype led to its loss during construction of the recombinant collection. Given that, and that association testing is greatly influenced by allele

frequencies, the high *petite* frequency *MIP1*-661A allele and other *petite*-influencing alleles of *CAT5*, *SAL1* and *MKT1* found in many S288c-derived strains [62] were not detected in our analyses. We did not explore the mechanisms of these novel *MIP1* alleles, though other alleles in *MIP1* are known to influence mtDNA stability [62,100,101]. Our findings suggest that additional naturally occurring *MIP1* variants contribute to basal differences in mtDNA stability in populations, though the frequencies of these alleles in yeast isolates have not been determined. Interestingly, *MIP1* expression was dependent on the genotypes of the mitonuclear associated genes, and *BNS1* expression was dependent on the genotype of *MIP1* (**S12 Table**). From this data, it can't be deduced whether cells alter *MIP1* expression in response to growth differences caused by genotypes of mitonuclear loci or if expression of mitonuclear genes is regulated by Mip1 levels. Significant mitonuclear genotype x *MIP1* genotype interactions influenced the expression of the mitonuclear associated genes *BNS1* and *YAT1* (**S12 Table**) and suggest that higher order interactions are involved in mtDNA stabilities.

We also found that *EST1* variants associated with mtDNA stability independently of mitotype and that its deletion altered *petite* frequencies. Est1 maintains telomerase at the linear ends of nuclear chromosomes [102,103] by stabilizing G-quadruplex secondary structures in the DNA [104]. G-quadruplexes play a role in stabilizing human mtDNAs and can cause mtDNA polymerase to stall [105]. The GC-clusters in yeast mtDNAs likely form secondary structures [72] though it is not known whether these form typical G-quad structures or whether they act to stabilize mtDNAs. While Est1 is normally targeted to the nucleus, a proteomics experiment found that Est1 coimmunoprecipitated with mitochondrial ribosomes just prior to cell division (in G2 phase) [106]. It is plausible that Est1 plays dual roles in nuclear and mitochondrial genome maintenance. Another non-specific mtDNA binding protein identified by our association model (but not verified here) was *ILV5*, a mitochondrial nucleoid associated protein that was previously shown to influence mtDNA stability [107]. While higher GC content typically stabilizes DNAs, we found that mtDNAs with the highest GC content were the least stable (**Fig 2A**). This is consistent with the observation that *petite* frequency differences between different yeast species correlates with their GC-cluster content [108]. Within *S. cerevisiae*, the M4 family of GC-clusters appears to be expanding in the mtDNAs with West African lineages by targeting another family of clusters [76] and likely explains the high *petite* frequencies in these strains (**Fig 2B**). Not much is known about functional differences between different categories of these mobile elements [72,75]. Previously, we carefully aligned 9 mtDNAs, including their long intergenic regions, and showed that only a small number of GC-clusters were in conserved positions [76]. Interestingly, the M4 clusters in the West African mtDNA in that alignment interrupted 15% (2 of 13) of the conserved clusters. The M4 clusters could destabilize mtDNAs directly or perhaps their expansion has interrupted genome stabilizing functions provided by the positions of the conserved clusters.

**Significance.** Strong nuclear effects likely mask important epistatic alleles in most mapping studies. The advanced intercrossed recombinant mapping population and association model presented here were specifically designed to detect mitonuclear epistasis, and allowed us to detect nuclear alleles that contribute to phenotypes in mitotype-specific ways, even in the presence of alleles with strong independent effects. We validated our approach by phenotyping mtDNA stability in this large collection and found that mitonuclear epistasis influencing *petite* frequencies mapped to genes involved in mitotic growth. Importantly, these statistical associations were validated in an independent genetic background, indicating that the statistical model identified interacting mitonuclear loci with physiological effects. The significance of these specific interactions in yeast populations is not yet known and will depend on allele frequencies and relative effect sizes in context with environments in mating populations. It is likely that the MNRC contains combinations of interacting loci that do not exist in nature,

although because naturally occurring genetic variation was used to create the mapping population, the pathways revealed should provide insight into evolutionary potential. The MNRC offers a powerful tool to identify mitonuclear interactions and helps us better understand and predict the complex genotype-phenotype relationships that shape life.

## Methods

### Yeast strains

All strains are listed in **S1 Table**. Wild yeast isolates, described in [109], were obtained from the National Collection of Yeast Cultures General. Deletion strains [110] were obtained from the Yeast Knockout Collection (Horizon Discovery). To replace mtDNAs, karyogamy deficient matings were performed as previously described [15].

### Media

Media recipes include: SD (6.7 g/L yeast nitrogen base without amino acids, 20 g/L glucose); CSM with or without amino acids as specified (SD + 800 mg/L CSM premix or as recommended by the manufacturer (Sunrise Science)); CSMEG (CSM + 30 mL/L ethanol and 30 mL/L glycerol instead of glucose); sporulation media (1% IOAc, 0,1% yeast extract, 0.05% dextrose); YPD (10 g/L yeast extract, 20 g/L peptone, 20 g/L glucose) supplemented with 10 mM CuSO4 when indicated; YPEG (YPD + 30 mL/L ethanol and 30 mL/L glycerol instead of glucose); YPDG (1% yeast extract, 2% peptone, 0.1% glucose, 3% glycerol). Sugar concentrations in CSM media were altered as indicated. For solid media, agar was added to 2% prior to autoclaving.

### *Petite* frequency assays

To assay rates of mtDNA deletions, 5 mL YPD cultures were inoculated with freshly grown colonies, grown in roller tubes at 30C for exactly 15.0 hrs, diluted and plated onto YPDG solid media. After 2–3 days at 30C, large (*grande*) and small (*petite*) colonies were counted manually or photographed and counted using an imaging system (sp-Imager-SA, S&P Robotics, Inc). To quantify *petite* frequencies in the deletion strains in **Fig 6**, 20 freshly grown colonies were scooped from solid media, diluted and plated directly onto YPDG solid media. Single assays were performed for each strain in the recombinant collections. All other assays were performed in 3–20 replicates. *Petite* frequencies are reported as the ratio of *petite* to total colonies ([#*petite* colonies/ total colonies] * 100).

### Growth phenotyping

Growth differences of RC strains were determined by spotting cells in high densities arrays onto solid CSM media using a BM3-BC colony processing robot (S&P Robotics) following the methodology described in [15] with 5–12 replicates per strain. The difference between maximum and minimum colony size formation at 30C was used as a proxy for growth. Growth rates of wild isolates were determined from liquid cultures in CSM media containing 0.2%, or 2.0% glucose at 30C or 37C. Strains were cultivated in 96 well microtiter plates using Biotek Eon photospectrometers using double orbital shaking. Optical densities (600nm) were recorded at 15-minute intervals until cells reached stationary phase. Maximal growth rates ($V_{max}$) were determined as the highest slope of regression lines modeled over sliding windows of 5 data points from growth curves using 4 replicates per strain.

## Multiparent recombinant strain collection

An F7 multiparent recombinant collection of *S. cerevisiae* yeast derived from 25 wild isolates and paired with 3 different mtDNAs was created for genome-wide association studies. We first created a set of parental strains containing a single mitotype with selectable markers to facilitate matings. We began with haploid derivatives (*MAT*a and *MATα ura3::KanMX)* representing 25 wild yeast isolates (**S1 Table**). To facilitate the selection of diploids between these haploid strains, an *arg8::URA3* disruption cassette from *BamHI* linearized plasmid, pSS1, was introduced into each *MATα* strain through chemical competence (EZ Yeast Transformation Kit (Zymo Research) or [111]) or electroporation [112]. Transformants (strains MLx2x1UA—MLx28x1UA) were selected on CSM-ura solid media and arginine auxotrophies and respiratory competencies were verified by replica plating to CMS-arg and YPEG, respectively. Correct integration of the *arg8::URA3* disruption cassette was verified through tetrad analysis: each *MATα arg8::URA3* strain was mated to their *MATa ura3::KanMX* isogenic counterpart, diploids were selected on SD media and sporulated on SPO media. Spores were dissected from ≥ 12 tetrads and printed to CSM-ura, CSM-arg, and YPEG media, verifying a 2:2 segregation of Arg- Ura+: Arg+ Ura- phenotypes. All crosses had >90% spore viabilities. The mtDNAs in the *MAT*a strains were replaced with the mtDNA from 273614N using karyogamy-deficient matings as previously described [15]. The mtDNAs from the *MATα arg8::URA3* strains were removed through ethidium bromide treatment.

To create the multiparent recombinant collection, the *MAT*a *ura3::KanMX* $\rho^+$ and *MATα ura3::KanMX arg8::URA3* $\rho^0$ strains were mated to create each possible heterozygous F1 diploid. To do this, aliquots (50 μL) of haploid strains were mixed with 200 μL fresh YPD media and incubated without mixing for 2 days at 30C in 96-well plates. Mating mixtures were harvested by centrifugation, washed (200 μL ddH$_2$0), re-suspended in 250 μL CSM-URA-ARG liquid media and incubated for 2 days at 30C with continuous shaking, and then spotted onto solid CSM-URA-ARG media. This yielded 279 F1 heterozygous diploids (of 284 attempted crosses). To create F1 haploids, 10 μL aliquots of diploid-enriched mixtures were spotted to sporulation media and incubated at room temperature until tetrads were visible via compound microscopy (7–11 days). The sporulated cells were collected and pooled by washing the plates with 5mL ddH$_2$0. To isolate spores from asci and vegetative cells, a random spore analysis protocol was followed [113], with modifications. Cell walls were digested by incubating 1 mL aliquots of the sporulated cells with zymolyase 20T (1 mg/ml) at room temperature for 1 hr with gentle rocking. The cells were centrifuged (12000 rpm, 4˚C, 1min), washed (1 mL ddH$_2$0) and resuspended in 100 μL ddH$_2$0. The treated cell mixture was vortexed vigorously for 3 minutes to adhere spores to tube walls. The supernatant was carefully aspirated, and the spores were gently washed (1mL ddH$_2$0) to remove remaining vegetative cells. To release spores from the plastic tube walls, cells were sonicated for 10–20 sec at 110 V (Stamina XP s50.0.7L, Sharper-Tek) in 1mL of 0.02% Triton-X. The released spores were centrifuged (12000 rpm, 4˚C, 1min), washed (1mL ddH$_2$0), and re-pelleted. The resulting freed spore mixtures were combined into a single tube with 1mL ddH$_2$0. Spore density was determined with a hemocytomer and plated for single F1 haploid colonies (~500 CFU/plate) onto 10 petri plates containing CSM-URA media (selecting for *MAT*a or *α ura3 arg8::URA3*) and 10 petri plates containing CMS-ARG media (selecting for *MAT*a or *α ura3 ARG8*). The F1 haploid cells were pooled by washing the colonies from each plate using ~5 mL ddH$_2$0. The pooled cells were washed in 5 mL ddH$_2$0, resuspended in 5mL YPD media and incubated at 30C without shaking. The random mating mixtures were washed in 5 mL ddH$_2$0, resuspended in 5 mL CSM-URA-ARG and incubated for 8–12 hours to enrich for F2 diploids. The cell mixtures were pelleted, resuspended in 2.5 mL ddH$_2$0, aliquoted (250 ul) onto solid sporulation media and incubated at room

temperature for 7 days. Spores were released and F2 haploids were isolated as described above. Random matings and haploid selection were repeated for a total of 7 meioses. Haploid F7 progeny were isolated as single colonies on YPD media, and replica plated to CSM-ARG, CSM-URA, and YPEG to determine auxotrophies and respiratory growth. Mating types were tested using mating type testers. Approximately 200 verified haploid recombinants were selected as strains for RC1 and include ~50 isolates of each selectable genotype. RC001-RC049: *MATa ura3 ARG8;* RC102-146: MATα *ura3 arg8*::URA3; RC201-247: *MATa ura3 arg8*::URA3; RC301-347: MATα *ura3 ARG8*.

The mtDNAs from each strain in RC1 were removed using an ethidium bromide treatment (creating RC001-347 $\rho^0$) and replaced using karyogamy deficient matings to create RC2 and RC3. RC2 strains contain the mtDNA from YPS606 and were created using mitochondrial donor strains SPK27 (for RC2:001–048), CK520E1 (for RC2:102–146 and RC2:301–347) and MKG109 (for RC2:301–307). RC3 strains contain the mtDNA from NCYC110 and were created using mtDNA donor strain TUC131.

### Genome sequencing and analysis

Genomic DNAs from 192 F7 recombinants from RC1 were isolated following Hoffman-Winston genomic DNA protocol (Hoffman & Winston 1987). DNA samples were concentrated using ZR-96 Genomic DNA Clean & Concentrator– 5 (Zymo Research Corp). DNA concentrations were measured using Qubit and diluted to the final concentration of 0.2 ng/μl. DNA sequencing libraries were prepared using an Illumina Nextera XT DNA Library Prep Kit according to manufacture instructions. The DNA concentration of each library was determined using Qubit, normalized, and pooled. The pooled libraries were sequenced using a single run of paired end 2 x 150 bp on Nextseq 500 (Illumina) at the Institute for Biotechnology and Life Sciences Technology at Cornell University. The reads for each RC strain were individually mapped to the annotated reference genome S288c (R64-2-1_20110203) using Bowtie2. Single nucleotide variants (SNPs) and regions of low and high coverage were identified using the Find Variations/SNPs function in Geneious v.8, filtering the results to regions with a minimum coverage $\geq 5$, minimum variant frequencies within the reads $\geq 0.8$ and the maximum variant $P$ values (the probability of a sequencing error) $\leq 10^{-6}$. To create a SNP table for association mapping, adjacent polymorphisms were merged, telomeric regions were removed and polymorphisms from each strain were combined into a single file. Low coverage areas were converted to deletions and the polymorphisms were filtered for biallelic variants with MAF >0.5%. This resulted in in 24,955 polymorphic sites.

### Statistical analysis and association testing

Statistics and plotting were performed using R 4.1.2 [114] and association models were performed using R 4.0.4 in RStudio [115]. The mtDNA sequences were analyzed for GC-cluster and intron content as previously described [76]. Accession numbers for mtDNAs are provided in **S4 Table**. Generalized linear models of the family binomial accounting for the different numbers of *petite* and *grande* colonies were performed using *glm* with the *cbind* function. ANOVAs for the analyses of growth or expression were performed using linear models with the *lm* function. Correlations were run using the *corr.test* function. Gene ontology classes for associated genes were identified using GO::TermFinder (Boyle, 2004 https://doi.org/10.1093/bioinformatics/bth456) and tested for over-representation using Fisher's exact tests.

### Genome-wide association analysis

A biallelic SNP table containing 24955 unique SNPs was used for association testing.

To address residual population structure in the recombinant strains, a principal component analysis was performed using PLINK v1.9 [116]. A simple association test was performed on the growth parameters from RC1 strains grown on YPD + 10 mM CuSO4 using the linear model: *colony size ~ covariates + nuclear SNP + error* where colony size is the strain mean derived from the linear model, nuclear SNP represented SNP variation at a given locus, and covariates included auxotrophies, mating type, and the first ten principal components from the PCA analysis. Association tests to detect mitonuclear interactions were performed using the *petite* frequency strain means of each strain in RC1, RC2 and RC3 in the following model: *petite frequency ~ covariates + nuclear SNP + mtDNA + nuclear SNP\*mtDNA+ error*. Covariates included *MIP1* alleles when indicated. False discovery rates were calculated using the "*qvalue*" package version 2.22.0 [117].

Associated SNPs that crossed FDR threshold were further filtered to non-synonymous SNPs within coding sequence (CDS) regions, and upstream regions within 250bp upstream of the CDS. Effect sizes for the associated SNPs with main effects were determined as the absolute differences in *petite* frequencies for strains with each variant weighted by their allele frequencies (effect size = $\Delta = |(\text{freq}_{allele1} \times petite \text{ freq}_{allele1}) - (f_{allele2} \times petite \text{ freq}_{allele2})|$).

## RT-qPCR

Cells from overnight CSM cultures were lysed according to [118] and total RNA was isolated using Qiagen RNeasy mini spin columns. RNA was quantified (Invitrogen Qubit Fluorometer) and diluted to 92 ng/μL. BioRad iScript cDNA Synthesis Kit was used to create cDNA, according to manufacturer's instructions. RT-qPCR was performed on a BioRad CFX Connect Real-Time PCR Detection System by adding 2 μl (~2.3 ng/μl) cDNA to 10 ul of BioRad SsoAdvanced Univerisal SYBR Green Supermix, 1 μL each of forward and reverse primers (500 nM) and 6 uL nuclease-free $H_2O$ in BioRad Hard-Shell 96-Well PCR Plates according to the following protocol: [95C-30s, (95C-15s, 60C 15-40s)$_{40}$, 65C-95C-0.5C/5s]. Primer sequences are provided in **S14 Table.** Relative expression was determined as the residuals from a linear regression of 1/Ct for each candidate gene against 1/Ct for the control gene, UBC6 as previously described [119]. This approach controls for differences in RNA extraction and cDNA manufacturing efficiencies and results in more normally distributed data where higher residual values correspond to higher starting mRNA levels.

## Supporting information

**S1 Table. Strain table.**
(XLSX)

**S2 Table. ANOVA *Petite* frequencies in wild isolates vary by strain and population.**
(XLSX)

**S3 Table. ANOVA mtDNAs influence *petite* frequencies in the Y55 nuclear background.**
(XLSX)

**S4 Table. mtDNA GC-clusters and introns.**
(XLSX)

**S5 Table. ANOVA *Petite* frequencies influenced by mitonuclear interactions in a 4x4 mitonuclear genotype panel.**
(XLSX)

**S6 Table. Recombinant Collection SNP descriptions.**
(XLSX)

**S7 Table. Associated SNPs that influence *petite* frequencies independent of mitotype (main effect loci).**
(XLSX)

**S8 Table. Associated SNPs that influence *petite* frequencies dependent on mitotype (mitonuclear loci).**
(XLSX)

**S9 Table. Effect sizes of main effect loci.**
(XLSX)

**S10 Table. Effect sizes of mitonuclear loci.**
(XLSX)

**S11 Table. ANOVAs Mitonuclear effects of *HGH1* and *YAT1*.**
(XLSX)

**S12 Table. ANOVA Mitonuclear effects of *HGH1*, *BSN1*, *SMI1*.**
(XLSX)

**S13 Table. ANOVAs Expression differences of mitonuclear candidate loci.**
(XLSX)

**S14 Table. Primer sequences for qRT-PCR.**
(XLSX)

**S15 Table. SNP allele frequencies in MNRC.**
(XLSX)

**S1 Fig. Strains harboring original mtDNAs have lower *petite* frequencies than those with synthetic mitonuclear combinations.** *Petite* frequencies of strains with the original (gold) vs. synthetic (grey) mitonuclear genotypes from **Fig 3** are replotted as box plots with the synthetic combinations combined. All nuclear and mtDNAs are from strains with West African lineages. ANOVA significances are shown. * $P<0.05$, ** $P \leq 0.005$, *** $P \leq 0.001$.
(TIF)

**S2 Fig. Association Mapping using the Recombinant Collection.** A Manhattan plot of -log10 of *P* values plotted against chromosomal position shows associations for maximal colony sizes for RC1 strains grown on copper sulfate. A single peak, corresponding to the location of *CUP1* on Chr. 8, is the only significant association at FDR<1%.
(TIF)

**S3 Fig. Strains with the GC-rich mtDNA in RC3 strains have higher *petite* frequencies.** *Petite* frequencies of strains from RC1, RC2, and RC3 are presented as violin plots. RC1 strains harbor mtDNA from 273614N containing a medium number (137) of GC-clusters. RC2 strains harbor mtDNA from YPS606 containing a low number (117) of GC-clusters. RC3 harbor mtDNA from NCYC110 containing a high number (210) of GC-clusters.
(TIF)

**S4 Fig. GWAS using MIP1 variants as covariates increases power to detect mitonuclear associations.** Manhattan plots show mitotype **A.** independent and **B.** mitotype dependent associations, when *MIP1* variants were included as covariates. Red lines indicate FDR thresholds at 0.1% for nuclear associations and 5.0% for mitonuclear associations.
(TIF)

**S5 Fig. Mitonuclear interactions of BNS1, SMI1 and HGH1 on *petite* frequencies.** Interaction plot follows *petite* frequencies for each nuclear genotype paired with different mtDNAs. See **S11 Table** for ANOVA. Data was collected using the same assay as performed for phenotyping the RCs (with 8–12 replicates) and cannot be combined with the data shown in **Fig 6**. (TIF)

**S6 Fig. *Petite* frequencies do not correlate with mRNA expression of associated genes.** Normalized expression levels (as residuals from regression lines of mRNA levels of each gene compared to a control gene) were plotted against *petite* frequencies. All genes showed positive correlation with *petite* frequencies, though no correlation was statistically significant. (TIF)

**S7 Fig. Expression of candidate genes by nuclear haplotype. A.** Normalized expression levels of each candidate gene separated by haplotypes of **A.** mitonuclear candidate loci or **B.** *MIP1* loci. The mitonuclear haplotypes represent the SNPs with highest effect sizes for each candidate gene. *P* values for significant differences are shown. All other comparisons were non-significant. (TIF)

**S8 Fig. Synthetic mitonuclear genotypes with increased *petite* frequencies have increased growth rates.** Maximum colony sizes for strains containing original or synthetic mitonuclear genotypes are presented as boxplots. Each synthetic mitonuclear genotype had higher growth, and higher *petite* frequencies (**S1 Fig**), than the original mitonuclear genotype. Growth data were from NGUYEN *et al.* 2020 and collected in the same conditions as the *petite* assays were performed. * $P<0.05$, ** $P \leq 0.005$, *** $P \leq 0.001$. (TIF)

**S9 Fig. *Petite* frequency of candidate genes associated with mtDNA stability.** *Petite* frequencies of strains containing deletions of candidate genes that did not depend on mitotype are shown as boxplots. Significant differences between the *petite* frequencies of the parental strain and each gene disruption, based on 3 replicates for each strain, are shown. Colors indicate chromosomal location of genes. * $P<0.05$, ** $P \leq 0.005$, *** $P \leq 0.001$. (TIF)

## Acknowledgments

We gratefully acknowledge T.D. Fox for the gift of plasmid, pSS1 and L.P. Musselman for assistance with RT-qPCR. We are also grateful to Binghamton University undergraduate students K. Dave, B. DeJesus, J. DeJesus, A. Federico, G. Lal, S. Sondhi, S. Oh, B.A. Wong, and A. Ziesel for technical assistance with *petite* frequencies in RC1.

## Author Contributions

**Conceptualization:** Kenneth Chiu, Anthony C. Fiumera, Heather L. Fiumera.

**Formal analysis:** Tuc H. M. Nguyen, Austen Tinz-Burdick, John F. Wolters, Anthony C. Fiumera, Heather L. Fiumera.

**Funding acquisition:** Kenneth Chiu, Anthony C. Fiumera, Heather L. Fiumera.

**Investigation:** Tuc H. M. Nguyen, Austen Tinz-Burdick, Meghan Lenhardt, Margaret Geertz, Franchesca Ramirez, Mark Schwartz, Michael Toledano, Brooke Bonney, Benjamin Gaebler, Weiwei Liu, John F. Wolters, Heather L. Fiumera.

**Supervision:** Heather L. Fiumera.

**Visualization:** Tuc H. M. Nguyen, Austen Tinz-Burdick, Heather L. Fiumera.

**Writing – original draft:** Tuc H. M. Nguyen, Anthony C. Fiumera, Heather L. Fiumera.

**Writing – review & editing:** Tuc H. M. Nguyen, Austen Tinz-Burdick, Meghan Lenhardt, Margaret Geertz, Franchesca Ramirez, Mark Schwartz, Michael Toledano, Brooke Bonney, Weiwei Liu, John F. Wolters, Kenneth Chiu, Heather L. Fiumera.

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
