## [Decision Letter · Decision Letter 0]

26 Oct 2022

Dear Dr. Fiumera,

Thank you very much for submitting your Research Article entitled 'Mapping mitonuclear epistasis using a novel recombinant yeast population' to PLOS Genetics. I apologize for the slow review process, but one of the reviewers needed extra time for his/her evaluation

Your manuscript was evaluated at the editorial level and by two independent peer reviewers. The two independent reviewers did not agree regarding whether your submitted work provides information to warrant publication in PLOS Genetics. Reviewer 1 does not support publication in the journal due to a lack of mechanistic discoveries; the reviewer also raises a number of technical issues. In contrast, Reviewer 2 finds your work to have uncovered new information and to have created a useful collection of mapping strains. Based on the reviews, we will not be able to accept this version of the manuscript, but we would be willing to review a much-revised version. We cannot, of course, promise publication at that time.

Should you decide to revise the manuscript for further consideration here, your revisions should address the specific points made by each reviewer.  In particular, address each of Reviewer 2's queries. Reviewer 1 has numerous concerns. We request that you address the novelty of your work and how your results relate to previous publications. Thus, regarding Reviewer 1's comment 4, please address whether you also discovered MKT1, SAL1, and CAT5. Regarding comment 1 on the use of the BY4741 background for your studies, we realize this is the only background available for the genome-wide approach and, moreover, it seems unlikely that deletion vs. allele variation would influence the results. Nevertheless, it would be worthwhile to discuss whether employment of the BY4741 could bias your results. Along the same lines (comments 2, 3, and 5), it would be valuable to verify your conclusions regarding growth rates by employing at least one additional strain. Finally, please  address each of the technical issues raised by Reviewer 1. We will require a detailed list of your responses to the review comments and a description of the changes you have made in the manuscript.

If you decide to revise the manuscript for further consideration at PLOS Genetics, please aim to resubmit within the next 60 days, unless it will take extra time to address the concerns of the reviewers, in which case we would appreciate an expected resubmission date by email to plosgenetics@plos.org.

We are sorry that we cannot be more positive about your manuscript at this stage. Please do not hesitate to contact us if you have any concerns or questions.

Yours sincerely,

Anita K. Hopper

Academic Editor

PLOS Genetics

Kirsten Bomblies

Section Editor

PLOS Genetics

Reviewer's Responses to Questions

**Comments to the Authors:**

Reviewer #1: please see attachment.

Reviewer #2: Nguyen et al. describe a genetic mapping experiment in yeast to identify mitochondrial, nuclear and mito x nuclear epistatic effects influencing phenotypes and mtDNA genome composition. The authors develop new mapping panels of yeast strains that are particularly focused on mapping mitonuclear interactions, the Mitonuclear Recombinant Collection (MRC). A focal phenotype of the study is the stability of the mitochondrial genome; deletions of mtDNA sequences are common among yeast strains and can be associated with growth phenotypes that are indicative of disease models as well as general yeast fitness.

A screen of natural isolates from around the world revealed significant variation in the incidence of the petite phenotype (small colonies in a growth assay), which have been, and were shown to be, associated with mtDNA deletions. Subsequent analyses in new stains pairing different mtDNAs with different nuclear backgrounds revealed that nuclear, mtDNA and mito x nuclear interactions explained the variation in growth phenotypes. To map the specific loci involved, the Mitonuclear Recombinant Collection was constructed among 25 nuclear chromosomal isolates engineered to have the same mtDNA, and allowed to recombine for 7 generations. Recombinant haploid descendants of this process were then paired with two alternative mtDNAs generating three different mapping panels harboring extensive nuclear polymorphism on alternative mtDNA backgrounds. Sequencing and phenotyping of these panels allowed gene mapping with GWAS approaches that could detect main and mitonuclear epistatic effects. A number of nuclear loci associated with growth (petite) and mtDNA deletion traits were identified, notably the mitochondrial DNA polymerase gene MIP1. In addition, the molecular aspects of the mtDNA deletions were attributable to GC rich motifs that contribute to deletions. The details of gene and mutation validation were reported.

Overall this panel of strains is a very powerful new mapping tool for mitochondrial genetics, but more generally for considering epistasis mapping questions. The text is clear and the statistical analyses seem fine, as there are strong signals that were followed up with empirical validation. This paper will be of general interest to the PLoS Genetics readership.

General Comments:

The different rank orders of QTL effects in the three mapping samples (RC1, 2, 3 - Figure 5) suggest that mtDNA affects the nuclear contribution to the petite frequency, or some kind of mitonuclear interaction. How can one be certain that the allele frequencies at nuclear loci are (precisely) the same in the different RC1, 2, 3 backgrounds? Any difference in these frequencies could alter the power to detect or discover a nuclear variant, which would be interpreted as a mitonuclear interaction when observed across RC1, 2, 3. Perhaps the breeding scheme for the MRC ensures this, but it seems there is plenty of opportunity for sampling drift among alleles in the final (and ongoing culture) of these panels.

How much of the mtDNA deletion effects were influenced by heteroplasmy level, vs. simple presence absence? This may have been missed in reviewing the manuscript, but was heteroplasmy quantified as relative copy number, percent heteroplasmy, or a +/- trait?

**Have all data underlying the figures and results presented in the manuscript been provided?**

Reviewer #1: Yes

Reviewer #2: Yes

PLOS authors have the option to publish the peer review history of their article (what does this mean?). If published, this will include your full peer review and any attached files.

Reviewer #1: No

Reviewer #2: **Yes: **David Rand

---

## [Decision Letter · Decision Letter 1]

13 Feb 2023

Dear Dr. Flumera,

Thank you very much for submitting the revised version of your Research Article entitled 'Mapping mitonuclear epistasis using a novel recombinant yeast population' to PLOS Genetics.

The manuscript was evaluated at the editorial level and by previous Reviewer 2 who was asked to comment on your responses to both Reviewers 1 and 2. The reviewer of your current version finds that you have justified conducting the study with wild-type yeast and that your new collection is valuable. Regarding your response to previous Reviewer 2, this reviewer requests that your further address whether recombinant nuclear haplotypes are identical across mtDNA backgrounds. 

We therefore ask you to modify the manuscript according to the review recommendations. Your revisions should address the specific points made by the reviewer.

Yours sincerely,

Anita K. Hopper

Academic Editor

PLOS Genetics

Kirsten Bomblies

Section Editor

PLOS Genetics

Reviewer's Responses to Questions

**Comments to the Authors:**

Reviewer #2: Nguyen et al. revision. The authors have provided a revised version of the manuscript with detailed responses to the questions from the reviewers. The comments from reviewer 1 were more extensive and expressed concern about the suitability of the paper for PLoS Genetics; those from reviewer 2 more limited and supportive of publication.

The response to reviewer 1 was sufficient based on clarifying the motivation to use wild strains as a source of natural alleles. The fact the petite frequencies are >5x higher in lab strains than in wild strains means the QTL approaches in the current study could well identify very different loci than in previous analyses. Moreover, the authors have done adequate functional validation of specific mitonuclear interactions to demonstrate meaningful fitness and physiological effects. The different philosophies of identifying novel genetic factors with a new panel, vs. identifying specific mechanisms of strong, lab-strain mutants has been well explained by the rebuttal statement. The value of the new collection is high and could lead to new genetic mechanisms of mitonuclear interactions. In addition, novel fermentation products could emerge in applied contexts.

The comments of reviewer 2 focused on potential spurious effects of allele frequencies differences between the recombinant populations that might leave a signal of QTLs for mitonuclear interactions. The statement that the collection of recombinant nuclear haplotypes is ‘identical’ across mtDNA backgrounds is taken on faith that the yeast genetic cytoplasmic replacement are clean. It would still help to address this by testing whether the evidence for nuclear x nuclear epistatic effects do not differ across the different mtDNA backgrounds. Any such variation would/could be statistically identified as a mitonuclear effect based on the 2-way anova approach. There certainly must be strong nuclear-nuclear epistatic effects on the phenotype measured (does the effect of nuclear SNP x depend on the state of nuclear SNP y elsewhere in the genome; this could be modeled with jackknife samples of the RC2 and RC3). If there are big differences between the population/samples in these effects that would requires some additional explanation in relation to the mtDNA epistases.

**Have all data underlying the figures and results presented in the manuscript been provided?**

Reviewer #2: Yes

PLOS authors have the option to publish the peer review history of their article (what does this mean?). If published, this will include your full peer review and any attached files.

Reviewer #2: **Yes: **David Rand

---

## [Editor Report · Decision Letter 2]

10 Mar 2023

Dear Dr Fiumera,

We are pleased to inform you that your manuscript entitled "Mapping mitonuclear epistasis using a novel recombinant yeast population" has been editorially accepted for publication in PLOS Genetics. Congratulations!

Yours sincerely,

Anita K. Hopper

Academic Editor

PLOS Genetics

Kirsten Bomblies

Section Editor

PLOS Genetics

Comments from the reviewers (if applicable):

**Data Deposition**

http://datadryad.org/submit?journalID=pgenetics&manu=PGENETICS-D-22-01006R2

**Press Queries**

---

## [Editor Report · Acceptance letter]

24 Mar 2023

PGENETICS-D-22-01006R2 

Mapping mitonuclear epistasis using a novel recombinant yeast population 

Dear Dr Fiumera, 

We are pleased to inform you that your manuscript entitled "Mapping mitonuclear epistasis using a novel recombinant yeast population" has been formally accepted for publication in PLOS Genetics! Your manuscript is now with our production department and you will be notified of the publication date in due course.

With kind regards,

Zsofia Freund

PLOS Genetics

On behalf of:
